# Utilization of Rice Husk Ash in the Preparation of Graphene-Oxide-Based Mesoporous Nanocomposites with Excellent Adsorption Performance

**DOI:** 10.3390/ma14051214

**Published:** 2021-03-04

**Authors:** Tzong-Horng Liou, Yuan Hao Liou

**Affiliations:** 1Department of Chemical Engineering, Ming Chi University of Technology, 84 Gungjuan Rd., Taishan, New Taipei 24301, Taiwan; luisliutw01@gmail.com; 2Battery Research Center of Green Energy, Ming Chi University of Technology, 84 Gungjuan Rd., Taishan, New Taipei 24301, Taiwan

**Keywords:** rice husk ash, graphene oxide, SBA-15, nanocomposite, sustainable utilization

## Abstract

Rice husk is an agricultural biomass waste. Burning rice husks in an oxygenic atmosphere releases thermal energy and produces ash that is rich in silica. Rice husk ash (RHA) can be used as a sustainable source of silica for producing high-value-added products. In this study, mesostructural graphene oxide (GO)/SBA-15, a graphene-based hybrid material, was synthesized from RHA. The materials are inspected by Fourier transform infrared spectrometer, Raman spectrometer, field-emission scanning electron microscopy, transmission electron microscopy, surface area analyzer, and X-ray diffraction analyzer. Studies have revealed that GO/SBA-15 possesses various oxygen functional groups that are helpful for dye adsorption. The material consisted of high pore volume of 0.901 cm^3^/g, wide pores of diameter 11.67 nm, and high surface area of 499 m^2^/g. Analysis of the methylene blue (MB) adsorption behavior of GO/SBA-15 composites revealed that their adsorption capacity depended on the gelation pH, GO content, adsorbent dosage, and initial dye (MB) concentration. The highest adsorption capacity of GO/SBA-15 was 632.9 mg/g. Furthermore, the adsorption isotherms and kinetics of GO/SBA-15 were investigated. This study demonstrated the great advantage of treated RHA and the potential of this material for use in organic dye adsorption.

## 1. Introduction

Globally, biomass energy is a vital renewable energy source [1]. Rice husks (RHs) are residual products from the processing of rice in mills and are considered agricultural waste. However, RHs are a highly efficient thermal energy source and can be used to produce electricity [2]. RHs mainly comprise organic matter and ash. Burning RHs produces RH ash (RHA) (approximately one-fifth), which is rich in silica (approximately 98 wt.%) [3]. Annually, more than 701 million tons of RH is generated. However, considerable quantities of RHA (approximately 140 million tons) are wasted [4]. Currently, only a small portion of RHA can be efficiently utilized, which creates a disposal problem. Recycling RHA as a bioresource is a sustainable method for developing a circular economy. RHA can be used for fabricating CO_2_ capture [5], cement-based materials [6], catalysts [7], silicon carbide ceramics [8], zeolites (ZSM-5) [9], adsorbent materials [10], fuel cells [11], and mesoporous molecular sieves—Mobil Composition of Matter (MCM) and Santa Barbara Amorphous (SBA) series [12].

Graphene and graphene oxide (GO) have attracted considerable attention for their applications—such as in biomedical materials, water purification, energy-related fields, sensing, and electronics [13]—because of their excellent mechanical, thermal, and electric properties. GO has high surface area and contains epoxy, hydroxyl, and carboxyl functional groups. These oxygen-containing functional groups enable GO to interact with organic and inorganic compounds, rendering GO a potential candidate for pollutant adsorption [14]. However, GO is highly hydrophilic in aqueous media, which results in GO aggregation and poses a filtration difficulty. These phenomena are not conducive to dye extraction applications [15]. However, GO modifications, such as the combination of GO with support material—such as zeolites [16], metal oxides (ZnO, SiO_2_, Fe_3_O_4_, etc.) [17], activated carbons [18], and biopolymers (e.g., chitosan and cellulose) [19]—can prevent GO aggregation and increase recovery after wastewater treatment. Among these support materials, SBA-15 is a widely used mesostructural material that provides a stable support structure and has hexagonally packed uniform porosity, adjustable pore size, and high surface area [20]. SBA-15 can facilitate excellent GO dispersion to increase adsorption capacity and ensure fast kinetics and recyclability. RHA contains abundant silica and is therefore suitable for the synthesis of SBA-15.

Methylene blue (MB) is an organic dye that is widely used in industries such as the textile, printing, and leather product industries [21]. However, organic dyes discharged from industries cause severe pollution of water bodies. Most dyes have complex aromatic structures and are hazardous to humans and animals. Methods such as adsorption, ion-exchange, membrane filtration, and chemical precipitation are used to eliminate dye from wastewater [22]. Among these techniques, adsorption is the most effective method because of its high efficiency, ease of operation, and low cost.

GO and reduced GO (rGO) materials have been widely employed as efficient adsorbents. However, few studies have focused on dye adsorption when using graphene composites as a potential adsorbent. Fan et al. [23] prepared magnetic β-cyclodextrin-chitosan/GO composites and determined that the highest adsorption capacity for MB was approximately 100 mg/g. Wang et al. [24] prepared rGO/TiO_2_ hybrids and revealed that microwave-assisted rGO/TiO_2_ had higher adsorption capacity than directly reduced rGO/TiO_2_. Heidarizad and Sengor [25] decorated GO with MgO particles and used MB as the adsorbate. They revealed that the highest adsorption capacity occurred at pH 11. A highest adsorption capacity of 68 mg/g for MB removal was achieved using GO-cellulose composite aerogels [26].

Previous studies have already reported the preparation of adsorbents from RH or RHA. However, there is a lack of information in the literature about GO-based silica composites prepared from RHA for obtaining high adsorption capacity materials. This lack in existing literature is a motivation for the present study. In this study, we proposed a sustainable GO/SBA-15 nanocomposite preparation method using RHA waste as the silica source. A silicate sodium solution was first obtained through alkali extraction of RHA. GO was prepared from graphite powder by using a modified Hummers method. Subsequently, GO/SBA-15 was synthesized using a silicate sodium precursor, a GO suspension, and a surfactant solution through hydrothermal treatment. Fourier transform infrared spectrometry (FTIR), Raman spectrometry, field-emission scanning electron microscopy (SEM), transmission electron microscopy (TEM), surface area analysis, and X-ray diffraction (XRD) were performed to characterize the physical and chemical properties of the GO/SBA-15 composites. Furthermore, the adsorption properties of the cationic dye MB on GO/SBA-15 composites were investigated. The adsorption capacity was monitored through ultraviolet-visible spectrophotometry. The effects of the gelation pH, GO content, dosage of adsorbent, and initial dye concentration on the adsorption capacity of the hybrid materials were observed. Finally, the isotherm and kinetics of adsorption data were estimated.

## 2. Materials and Methods

### 2.1. Materials

High-purity graphite powder, tetraethyl orthosilicate (TEOS), and pluronic triblock copolymer (P123) were purchased from Sigma-Aldrich (Taufkirchen, Germany). Sodium nitrate, MB, hydrogen peroxide, and potassium permanganate were purchased from Acros Organics (Morris Plains, NJ, USA). Sulfuric acid, sodium hydroxide, and hydrochloric acid were obtained from Merck (Gernsheim, Germany). High-purity (99.995%) nitrogen and air were obtained from Sun Fu (Taipei, Taiwan).

### 2.2. Extraction of Silica from RHA

The elements and basic components of RHs were reported in our previous study [27]. Silica was extracted from PRA as follows [28]. RH samples were gathered from a rice plant. The samples were first washed with distilled water and then placed in a flask. A hot HCl solution was added to remove metallic impurities [29]. Acid-leached RHs were dried and then moved to a tubular furnace. The specimens were carbonized at 700 °C for 1 h by using highly purified nitrogen gas and then placed in a flask containing 1.5 M NaOH solution for extracting RHA. Sodium silicate solution was obtained by boiling the mixture at 100 °C for 1 h. Carbon solids were separated using a high-speed spinning centrifuge (Eppendorf Centrifuge 5702, Wesseling, Germany). Traces of metal and carbon residues were removed by using a membrane filter (Whatman Plc, Kent, UK). Filtration was performed several times before obtaining the final product.

### 2.3. Synthesis of GO

A modified Hummers method was used to prepare GO from graphite powder [30,31]. The graphite was first oxidized by treating a mixture of graphite powder, NaNO_3_, H_2_SO_4_, and KMnO_4_. The temperature of the mixture was maintained at 30–40 °C with constant stirring. After 2 h, water and H_2_O_2_ were added. The suspension was then washed using a diluted HCl solution. GO was collected through filtration and water-washing processes. Finally, the precipitate was dispersed in deionized water. Ultrasonication was then performed to obtain a uniform and stable GO suspension.

### 2.4. Preparation of RH-GO/SBA-15

RH-GO/SBA-15 was prepared by heating the surfactant mixture using the hydrothermal method as follows [32]: 1.0 g of P123, 6.0 g of 0.5 wt.% GO suspension, and 125 mL of 2.0 M HCl were mixed in a glass flask under continuous stirring at 35 °C. Then, 25 mL of sodium silicate solution (extracted from RHA) was slowly added to the surfactant solution. The pH of the solution was fixed at 2 and was measured using a pH meter (Metter Toledo, model S20-K, Greifensee, Zürich, Switzerland). The mixture was agitated for 1 day and then moved to a Teflon-lined autoclave (Fu Chang Company, Tao Yuan City, Taiwan). The mixture was heated to 100 °C for 1 day. The suspension was then cooled, and the obtained specimen was rinsed using distilled water until the final solution was pH neutral. After drying the solid in an air oven, the as-synthesized sample was purged using nitrogen gas and then placed in a tubular furnace and heated to 550 °C for 6 h. The collected final product was RH-GO/SBA-15. RH-SBA-15 and T-SBA-15 (pure SBA-15 material) were obtained from RHA and commercial tetraethyl orthosilicate (TEOS), respectively, by using the same hydrothermal process and without GO addition.

The same hydrothermal process was followed when adding various quantities of GO suspension to the surfactant solution. Here, 1.5, 3.0, 4.5, and 6.0 g of GO suspension were added, and the corresponding samples were labeled RGS-1.5, RGS-3.0, RGS-4.5, and RGS-6.0, respectively.

### 2.5. Characterization of the Hybrid Materials

A N_2_ adsorption-desorption experiment was performed using a Micrometrics ASAP 2020 (Norcross, GA, USA) analyzer to investigate the pore characteristics of T-SBA-15, RH-SBA-15, and RH-GO/SBA-15 specimens. The functional groups of the specimens were examined using a Shimadzu FTIR-8300 analyzer (Nakagyo-ku, Kyoto, Japan). The diffraction patterns of the mesophase were collected on an X’pert pro system X-ray diffractometer (PANalytical, Malvern, UK) using Cu-kα radiation. The mesostructures of the samples were examined using a JEM-2100 (JEOL, Akishima, Tokyo, Japan) transmission electron microscope, and their chemical elements were inspected through energy-dispersive X-ray spectroscopy (EDS). The morphologies of the samples were recorded using a S-3400N (HITCHI, Chiyoda, Tokyo, Japan) field-emission scanning electron microscope. Raman spectra of the samples were obtained using 632-nm He–Ne laser excitation and a confocal micro-Renishaw Raman spectrometer (Renishaw, Gloucestershire, UK). The zeta potentials were measured by a zeta potential instrument (Zetasizer 2000, Worcestershire, UK). The pH values in the range of 2–10 were adjusted by dilute hydrochloric acid or sodium hydroxide aqueous solutions.

### 2.6. Adsorption Experiment

The adsorption behavior of the adsorbent samples was investigated using a batch method. MB aqueous solutions of concentrations 20, 30, 40, and 50 mg/L were used for the adsorption study. The pH of the solution was fixed at 7. The initial pH adjustments were carried out either by hydrochloric acid or sodium hydroxide solutions. Digital pH meter (Metter Toledo, model S20-K, Greifensee, Switzerland) was used for the pH measurements. Here, 10 mg of the RH-GO/SBA-15 sample was immersed in 250 mL of the MB dye solution. The effects of the gelation pH, GO content, adsorbent dosage, and initial dye concentration on the adsorption capacity of RH-GO/SBA-15 were determined by varying the gelation pH from 2 to 6, GO suspension from 1.5 to 6.0 g, and adsorbent dosage from 5 to 30 mg. Residual dye solutions were collected at regular intervals between 1 and 300 min and purified though centrifugation and membrane filtration. An ultraviolet-visible Genesys spectrophotometer (Thermo Electron Corporation, Waltham, MA, USA) was used to observe the change in concentration of the residual dye solution. The maximum adsorption wavelength was 665 nm. The adsorption capacity of RH-GO/SBA-15 (*q_t_*, mg/g) was estimated using the following mass balanced equation:(1)$$q_{t}=\frac{\left(C_{o}-C_{t}\right)V}{W}$$
where *W* (g) is the mass of RH-GO/SBA-15, *V* (mL) is the volume of the MB dye, and *C**_o_* (mg/L) and *C**_t_* (mg/L) are the initial MB concentration and MB concentration at time *t*, respectively.

## 3. Results and Discussion

### 3.1. Analysis of Fundamental Composition and Phase Properties

The low-angle XRD patterns of RH-GO/SBA-15 powders for various gelation pH values are displayed in Figure 1a. At pH 2, the spectrum of the sample exhibited three peaks at the (100), (110), and (200) planes, which corresponded to the typical characteristics of the hexagonal symmetrical SBA-15 structure [33]. The (110) and (200) peaks at pH 3–6 were considerably weaker, which indicated degradation of the SBA-15 mesostructure. A strongly acidic condition favored the formation of a highly ordered mesoporous material. Figure 1b presents the XRD patterns of RH-GO/SBA-15 synthesized using various quantities of the GO suspension. The four spectra exhibit the same three reflection planes at (100), (110), and (200), which indicate that the pore framework did not change during the chemical combination of GO with RH-SBA-15. However, the relative intensity of the peaks decreased with an increase in the quantity of GO. This occurred because the combination of GO with SBA-15 probably caused slight obstruction of silica pores [32]. The RH-SBA-15 and T-SBA-15 samples in Figure 1b also exhibited three peaks at the (100), (110), and (200) planes, indicating the formation of SBA-15 structure. SBA-15 material obtained from RHA shows a stronger relative peak intensity than material obtained from TEOS.

The FTIR spectra of the RH and RHA samples are displayed in Figure 2a. The RH spectrum consists of peaks corresponding to various functional groups: OH (~3400 cm^−1^), C–H (~2900 cm^−1^), C=O (~1692 cm^−1^), C=C (~1615 cm^−1^), –CH_2_ (~1492 cm^−1^), C–O (~1158 cm^−1^), and R–OH (~1051 cm^−1^) [34]. RHA was obtained from the combustion of RHs in air. Several bands in the spectrum of RH were not present in the spectrum of RHA because of vaporization of organic substances in the RHs. In the spectrum of the RH-GO/SBA-15 sample, presented in Figure 2b, the band at 3200–3700 cm^−1^ corresponds to the O–H hydroxyl groups. The 1635 cm^−1^ peak results from the presence of C=O vibrations. The peak corresponding to C=C groups is close to 1605 cm^−1^. The peak at 1375 cm^−1^ is a result of -CH_3_ stretching vibrations. The Si–OH groups are represented by the peak at 980 cm^−1^. The peaks at approximately 790 and 450 cm^−1^ correspond to Si–O–Si vibrations [35]. The results of the FTIR analysis revealed that RH-GO/SBA-15 contained carboxylic, carbonyl, and hydroxyl groups [36]. The high negative potential of oxygen-containing functional groups is conducive to the adsorption of cationic MB molecules.

Figure 3 shows the Raman spectra of the RH-SBA-15 and RH-GO/SBA-15 samples. No peaks are present in the spectrum of RH-SBA-15 (Figure 3a). For the RH-GO/SBA-15 material, the spectrum has strong G and D bands at approximately 1588 and 1351 cm^−1^, respectively (Figure 3b). Table 1 lists the peak intensity and I_D_/I_G_ ratio, which ranges between 0.9664 and 0.9760. The I_D_/I_G_ ratio slightly increased with an increase in the amount of GO suspension from 1.5 to 6.0 g. The RH-GO/SBA-15 samples had high I_D_/I_G_ ratio probably because they contained more oxygen functional groups and vacancy defects than the pure RH-SBA-15 samples. This observation indicated that all RH-GO/SBA-15 samples had a high degree of oxidation [35].

Figure 4 shows the effect of pH on the zeta potentials of the RH-GO/SBA-15 composite. It is found that the zeta potentials of sample are highly negative in the pH range between 2 and 10. This observation indicated that adsorbents disperse homogeneously and resist aggregation. MB ions were those of a cationic dye, which existed as positively charged ions in aqueous solution. An increase in OH− at high pH could enhance MB adsorption due to the interaction of electrostatic attraction [27].

### 3.2. Surface Morphology of RH and Mesoporous Silica Samples

Figure 5a shows an SEM image of the outer epidermis of a RH. The husk was swollen, and its tissues were regularly arrayed. Figure 5b displays an SEM image of the inner surface of the RH. The surface of the sample was rough, and agglomeration of rectangular tissues can be observed. The silica in the RH is mainly centralized on the outer epidermal tissue [34]. Figure 5c displays an SEM image of RH-SBA-15 obtained without the addition of GO and reveals that the sample was composed of small particles that resembled identical coils. The size of the particles was approximately 0.55 μm. Figure 5d–f displays the RH-GO/SBA-15 samples for various GO contents. The GO suspension content was varied from 1.5 to 6.0 g. RH-GO/SBA-15 exhibited similar surface morphology to RH-SBA-15 (a group of micrometer-size particles). No additional GO appeared on the exterior of SBA-15, which indicated that the GO mixed well with the SBA-15 particles.

The mesostructures of the prepared RH-SBA-15 and RH-GO/SBA-15 were characterized through TEM and EDS (Figure 6). Figure 6a confirms the existence of a well-ordered array of 11.8-nm-diameter homogeneous pores and parallel channels in the mesoporous RH-SBA-15. The silica sample exhibited a coil appearance, which was consistent with the SEM images (Figure 5). Figure 6b displays the large-scale hexagonal mesostructure of the RH-SBA-15 sample. The hexagonal mesostructure resulted from the electrostatic interaction between silicate species and the self-assembly surfactant [37]. Figure 6c illustrates the parallel channels of pores and hexagonal structure of the RH-GO/SBA-15. This image demonstrates that the addition of GO particles did not destroy the well-organized structure of the SBA-15. A dark shadow is present on the surface of the mesoporous channels in the image, indicating that RH-SBA-15 powder was encapsulated by the GO flakes, which concurred with the observation of Li et al. [38]. Figure 6d shows the sheet-like GO flakes located on the exterior of silica pores. The wrinkled GO layer confirmed that the GO had high specific surface area and adsorption capacity. The EDS spectrum of RH-GO/SBA-15 (Figure 6e) reveals the presence of carbon, oxygen, and silicon, providing further proof that SiO_2_ and GO were the major components.

### 3.3. Pore Structure Analysis

The nitrogen desorption isotherm of T-SBA-15 sample (Figure 7a) exhibits a type IV hysteresis loop. When the P/Po ratio was 0.6–0.8, the isotherm had a narrow hysteresis loop. This observation indicated the formation of relatively uniform mesopores [32]. The pore size in T-SBA-15 was approximately 7.78 nm, as displayed in Figure 7b. Figure 7c indicates that the isotherms of both RH-SBA-15 and RH-GO/SBA-15 exhibit a type IV hysteresis loop, indicating the formation of mesopores [39]. The outline of the two isotherms is similar, indicating that the addition of GO particles did not change the mesoporous framework of SBA-15. Figure 7d illustrates that the pore size of RH-SBA-15 (11.73 nm) was slightly greater than that of RH-GO/SBA-15 (11.67 nm). As presented in Table 2, the surface area of RH-SBA-15 (656 m^2^/g) was larger than that of T-SBA-15 (625 m^2^/g). Therefore, RHA was an excellent sustainable source for synthesis of SBA-15. In addition, the T-SBA-15 sample had the smallest pores (7.78 nm) of all the materials, indicating that the T-SBA-15 sample exhibited high mass transfer resistance regarding the adsorbate diffusing into the pores of the adsorbent. The surface area and pore volume of RH-GO/SBA-15 were 499 m^2^/g and 0.901 cm^3^/g, respectively. The RH-GO/SBA-15 sample had lower surface area and pore volume than RH-SBA-15. GO is speculated to have filled the pores of SBA-15.

### 3.4. Dye Adsorption Study

Figure 8 illustrates the dye adsorption capacity of RH-GO/SBA-15, RH-SBA-15, GO, and T-SBA-15. The mesoporous silica obtained from RHA (RH-SBA-15 sample) exhibited higher adsorption capacity than the material obtained from commercial TEOS (T-SBA-15 sample), which implied that RH-SBA-15 had larger pores (Table 2) and, consequently, the MB diffusion rate in the interior of RH-SBA-15 was higher than that in T-SBA-15. Moreover, the RH-GO/SBA-15 sample exhibited a higher capacity for removing MB than did pure GO. The reduction in adsorption capacity of the pure GO can be attributed to GO aggregation in aqueous media. The equilibrium time of RH-GO/SBA-15 and RH-SBA-15 was approximately 60 and 120 min, respectively. The equilibrium time of RH-SBA-15 was two times that of RH-GO/SBA-15. The result also revealed that RH-GO/SBA-15 had higher MB adsorption ability than RH-SBA-15. The greater adsorption capacity of the RH-GO/SBA-15 sample could be attributed to the van der Waals, electrostatic, and π–π binding interactions resulting from dye–adsorbent attachment. MB ions are cationic dyes. Typically, GO is favorable for the adsorption of cationic dyes. In the present study, favorable electrostatic interaction was achieved between the dye and negatively charged RH-GO/SBA-15 surface resulting from oxygen-containing functional groups [14]. The contact time curves reveal that the adsorption of MB was rapid initially and then slowed near the equilibrium, which indicated that numerous adsorption sites were unoccupied during the initial adsorption period. When the adsorption sites of the adsorbent became almost saturated, the MB adsorption capacity reached a plateau and equilibrium was achieved, which indicated a decrease in the number of vacant adsorption sites.

The gelation pH plays a crucial role in the synthesis of the mesoporous framework of SBA-15 because surfactant micelles form at the optimum pH. Figure 9 shows that the adsorption capacity decreased with an increase in the pH from 2 to 6. The adsorption capacity was highest (456.63 mg/g) at pH 2 and lowest (298.66 mg/g) at pH 6. XRD analysis (Figure 1a) indicated that the gelation pH affected the pore tissue of RH-GO/SBA-15. In a strongly acidic environment (pH 2), the formation of the hexagonal framework was mainly attributed to the micelle-silica interaction [40]. The adsorbent exhibited highly ordered channels with a mesoporous structure, which favored delivery of the dye into the interior of RH-GO/SBA-15 but was unfavorable to the formation of surfactant micelles when the pH was higher than 3. The reason for the decrease in MB adsorption may have been degradation of the pore structure. To prevent such degradation, RH-GO/SBA-15 was synthesized at pH 2 in subsequent adsorption experiments.

To determine the influence of the GO content on the adsorption of MB, the amount of GO suspension used was varied from 1.5 to 6.0 g. Figure 10 indicates that an increase in the GO content led to higher MB adsorption capacity. The adsorption capacity was highest when the amount of GO suspension used was 6.0 g. Thus, high GO content was favorable to MB adsorption. The presence of more numerous active sites resulted in stronger interaction with the adsorbate. Ren et al. [26] investigated the adsorption of MB on GO/cellulose and revealed that the adsorption capacity of the material increased with an increase in the GO content, which concurs with our experimental results.

Figure 11 plots the contact time and adsorption capacity of the adsorbent for various initial concentrations of the dye. The MB concentration was varied from 20 to 50 mg/L. The adsorption capacity was highly dependent on the initial dye concentration. The MB adsorption capacity increased with an increase in the MB concentration. An increase in the MB concentration increased the concentration gradient and accelerated the diffusion of dye molecules into the adsorbent. The same conclusion for the adsorption of MB on magnetic GO was also investigated by Othman et al. [41].

Figure 12 indicates that the dosage of the adsorbent affected the degree of MB adsorption considerably. The adsorption capacity decreased from 510.97 to 226.44 mg/g when the adsorbent dosage was increased from 5 to 30 mg. The higher the adsorbent dosage, the higher the surface area and more adsorption sites were available. However, the decrease in the adsorption capacity for a high adsorbent dosage could be attributed to the adsorption sites remaining unsaturated, which resulted from the increase in the number of available adsorption sites. A similar observation was reported for the capture of chromium (III) ions on zeolite A when using raw kaolin as the silica source [42].

The maximum adsorption capacity of various GO-containing adsorbents reported in the literature is listed in Table 3 [18,43,44,45,46,47]. The RH-GO/SBA-15 composite was discovered to exhibit superior adsorption ability compared with these other materials. Thus, RHA can be used to obtain highly valuable RH-GO/SBA-15 nanomaterials and reduce wastewater pollution.

### 3.5. Adsorption Isotherm Experiments

Both the Langmuir and Freundlich models were applied to assess the adsorption system. The experiment was conducted using 10 mg of RH-SBA-15 or RH-GO/SBA-15 added to 50 mL of MB solution. The mixture was then agitated for 24 h. The MB concentrations were between 10 and 50 mg/L.

The linear Langmuir and Freundlich models can be expressed as follows:(2)$$\frac{1}{q_{e}}=\frac{1}{q_{L}}+\frac{1}{q_{L}K_{L}C_{e}}$$
(3)$$\log q_{e}=\log K_{F}+\frac{1}{n}\log C_{e}$$
where *C_e_* (mg/L) and *q_e_* (mg/g) are the concentration and adsorption capacity, respectively, at equilibrium; *q_L_* (mg/g) is the maximum adsorption capacity; *K_F_* and *n* are the Freundlich constants; and *K_L_* (mL/mg) is the Langmuir constant.

Figure 13 displays the Langmuir and Freundlich isotherm plots. The results for the isotherm parameters are presented in Table 4. For the RH-SBA-15 sample, the Freundlich isotherm model exhibited a relatively high R^2^, which indicated that the adsorption data corresponded to the Freundlich model. However, for RH-GO/SBA-15, the Langmuir equilibrium model was most suitable. The RH-GO/SBA-15 surface is a monolayer covered by MB molecules. The Langmuir constant, *q_L_*, which reflects the maximum adsorption capacity, was 478.47 and 632.91 mg/g for RH-SBA-15 and RH-GO/SBA-15, respectively. The *n* values of the two silica samples were higher than 1, indicating a favorable adsorption process [48]. The *R*_L_ of the two silica samples was between 0 and 1, which gives information about the adsorption process is more likely to occur [49].

### 3.6. Kinetic Studies

Kinetics experiments were performed to determine the influence of the contact time on adsorption and thus obtain the kinetics parameters. Adsorption kinetics were analyzed using the pseudo-first-order, pseudo-second-order, and intraparticle diffusion models [50]. The three models are governed by the following three equations:(4)$$q_{t}=q_{e}\left(1-e^{-k_{1}t}\right)$$
(5)$$\frac{t}{q_{t}}=\frac{1}{k_{2}q_{e}^{2}}+\frac{t}{q_{e}}$$
(6)$$q_{t}=k_{i}t^{0.5}+I$$
where *q_e_* and *q_t_* (mg/g) are the adsorption quantity of MB at equilibrium and time *t*, respectively; *k*_1_ and *k*_2_ are the constants of the pseudo-first-order and pseudo-second-order models, respectively; and *k_i_* is a constant of the intraparticle diffusion model.

The plots of adsorption capacity versus time when using the pseudo-first-order and pseudo-second-order models are displayed in Figure 14. Table 5 details the calculated kinetics parameters. For both RH-SBA-15 and RH-GO/SBA-15, the *R*^2^ indicated that the pseudo-second-order model had the best fit to the experimental data, indicating that the MB adsorption on the two adsorbents was primarily controlled by chemisorption. For the RH-GO/SBA-15 sample, the experimental *q_e_* (481.18 mg/g) was in favorable agreement with the calculated *q_e_* value (490.20 mg/g). For the intraparticle diffusion model, Figure 15 displays the plot of *q_t_* versus *t*^1/2^. The calculated parameters (*k_i_* and *C*) are listed in Table 5. For both RH-SBA-15 and RH-GO/SBA-15, the existence of two linearity characteristics indicates that the adsorption process involved two diffusion pathways. The first pathway was the transportation of the dye from the bulk solution to the surface of the adsorbent particles. The second pathway was MB penetration into mesopores and the adsorption of MB onto active sites. The lower *k_i_*_2_ value in the second step implied that final pore diffusion was the rate-limiting step. This observation was in agreement with the reported study on adsorption of anionic dyes onto hollow PCP-NH_2_ microcapsules [51].

## 4. Conclusions

RHs can be a sustainable resource. After burning RH into bio-energy, the residual RHA can be used for obtaining high-quality nanoproducts. In the present study, we developed mesoporous nanocomposites by combining GO and SBA-15. GO was obtained using a modified Hummers method, and SBA-15 was synthesized from RHA agricultural waste. The performance of the materials was evaluated using several adsorption experiments under various conditions. FTIR spectra indicated that RH-SBA-15 was chemically bonded to the GO flakes. TEM images revealed that the RH-SBA-15 surface was homogeneously covered with GO flakes. A strongly acidic environment favored the formation of RH-GO/SBA-15 with a highly ordered mesostructure. The pore framework of RH-GO/SBA-15 was extremely stable and not damaged by adding GO to the pores of SBA-15. The RH-GO/SBA-15 nanocomposite exhibited excellent MB adsorption capacity. The adsorption efficiency of the materials was in the following order: T-SBA-15 < RH-SBA-15 < RH-GO/SBA-15. Adsorption isotherms and kinetics studies revealed that the adsorption behavior of RH-GO/SBA-15 followed the Langmuir and pseudo-second-order models. The surface of RH-GO/SBA-15 had monolayer coverage of MB. The results could be useful for the development of adsorbents with excellent adsorption performance to be employed in the purification of dye wastewater.

## Figures and Tables

**Figure 1 materials-14-01214-f001:**
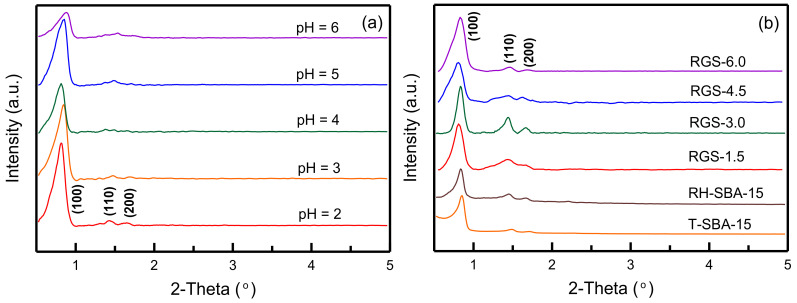
XRD patterns with low-angle diffraction of RH-GO/SBA-15: (**a**) different gelation pH values, and (**b**) different amounts of GO suspension.

**Figure 2 materials-14-01214-f002:**
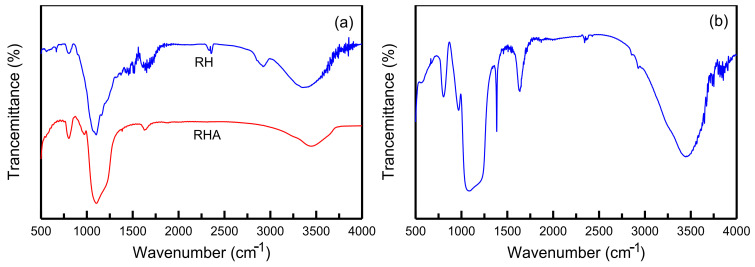
FTIR spectra: (**a**) RH and RHA, and (**b**) RH-GO/SBA-15.

**Figure 3 materials-14-01214-f003:**
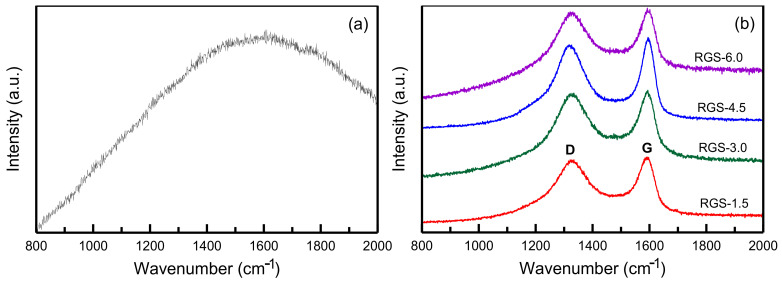
Raman spectra: (**a**) RH-SBA-15, and (**b**) RH-GO/SBA-15 for different amounts of GO suspension.

**Figure 4 materials-14-01214-f004:**
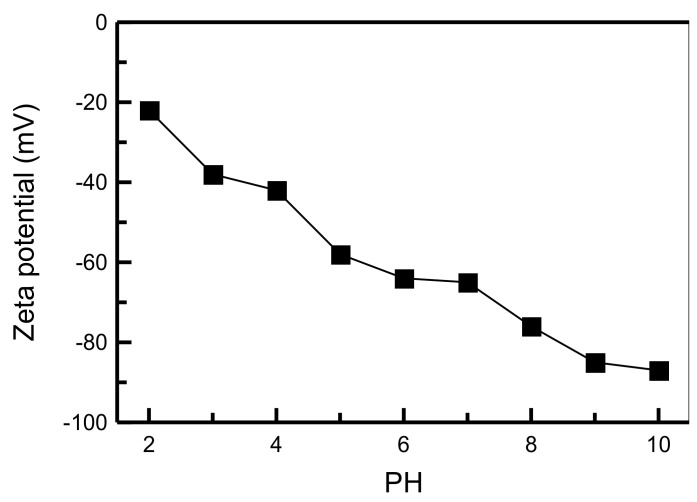
The pH dependence of zeta potentials of the RH-GO/SBA-15.

**Figure 5 materials-14-01214-f005:**
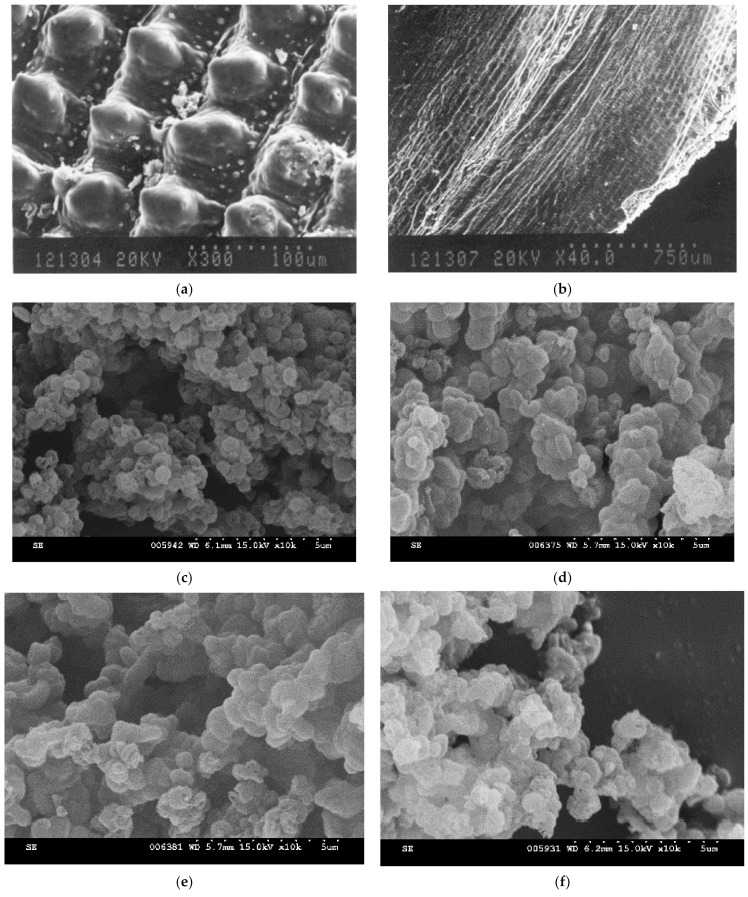
FE-SEM images: (**a**) outer epidermis of RH, (**b**) inner surface of RH, (**c**) RH-SBA-15, and different amounts of GO suspension (**d**) 1.5 g, (**e**) 3.0 g and (**f**) 6.0 g.

**Figure 6 materials-14-01214-f006:**
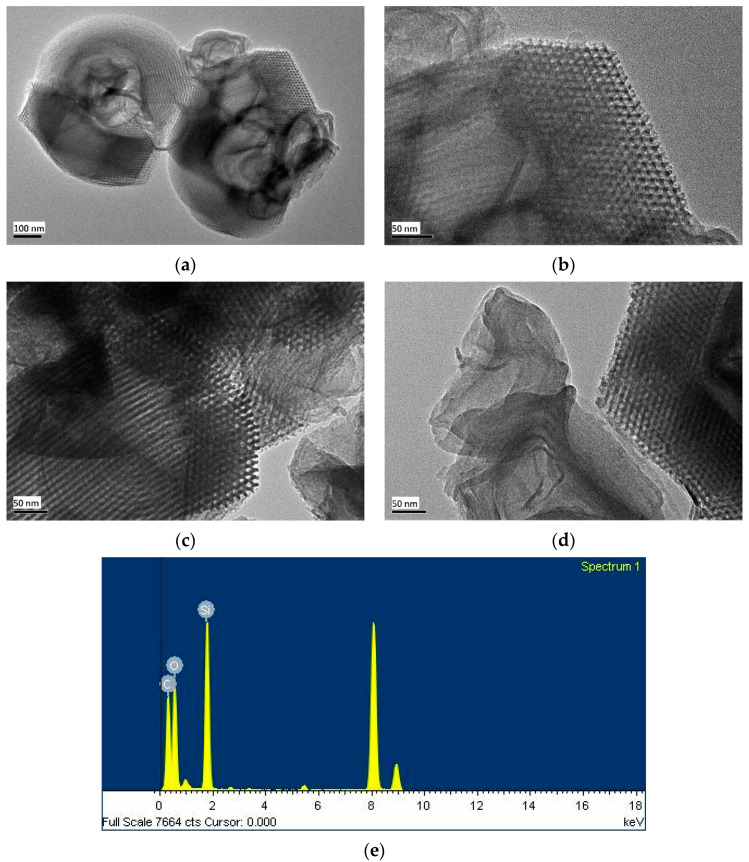
TEM pictures: (**a**,**b**) RH-SBA-15; (**c**,**d**) RH-GO/SBA-15; (**e**) EDS of RH-GO/SBA-15.

**Figure 7 materials-14-01214-f007:**
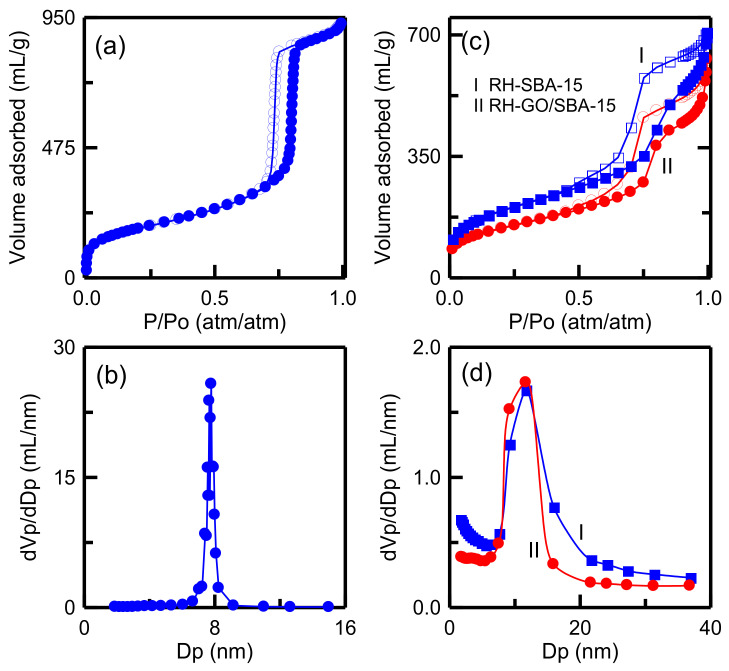
Nitrogen sorption isotherms and pore size distributions: (**a**,**b**) T-SBA-15; (**c**,**d**) RH-SBA-15 and RH-GO/SBA-15.

**Figure 8 materials-14-01214-f008:**
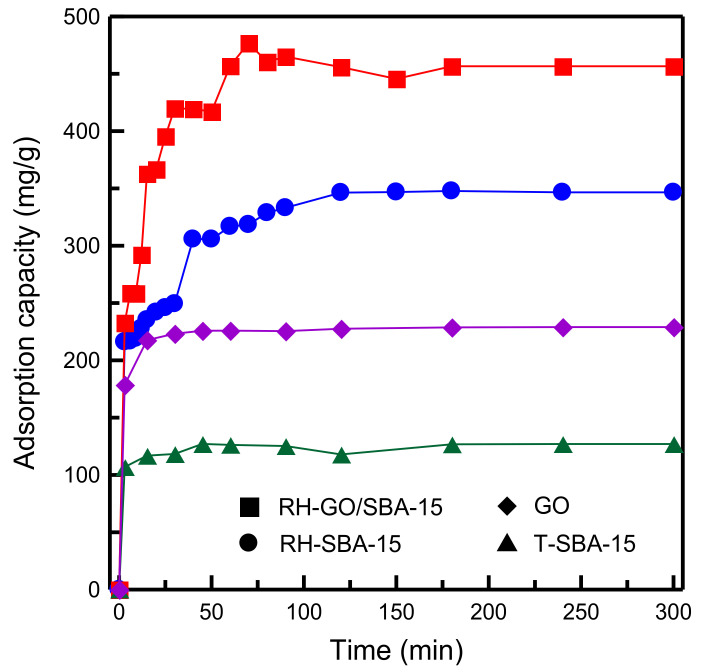
Comparison of adsorption capacity of GO, T-SBA-15, RH-SBA-15, and RH-GO/SBA-15.

**Figure 9 materials-14-01214-f009:**
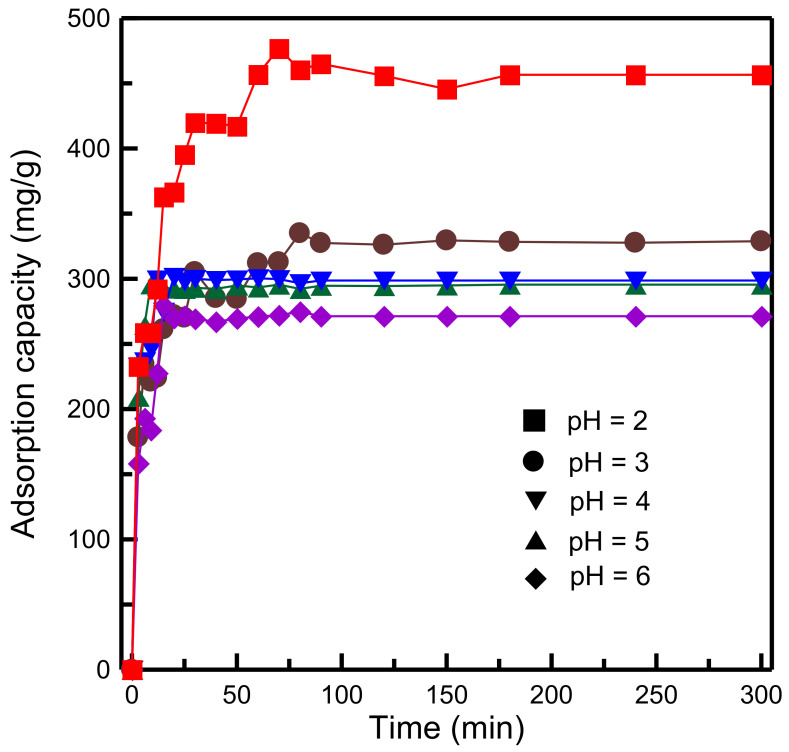
Influence of gelation pH values on adsorption capacity of RH-GO/SBA-15.

**Figure 10 materials-14-01214-f010:**
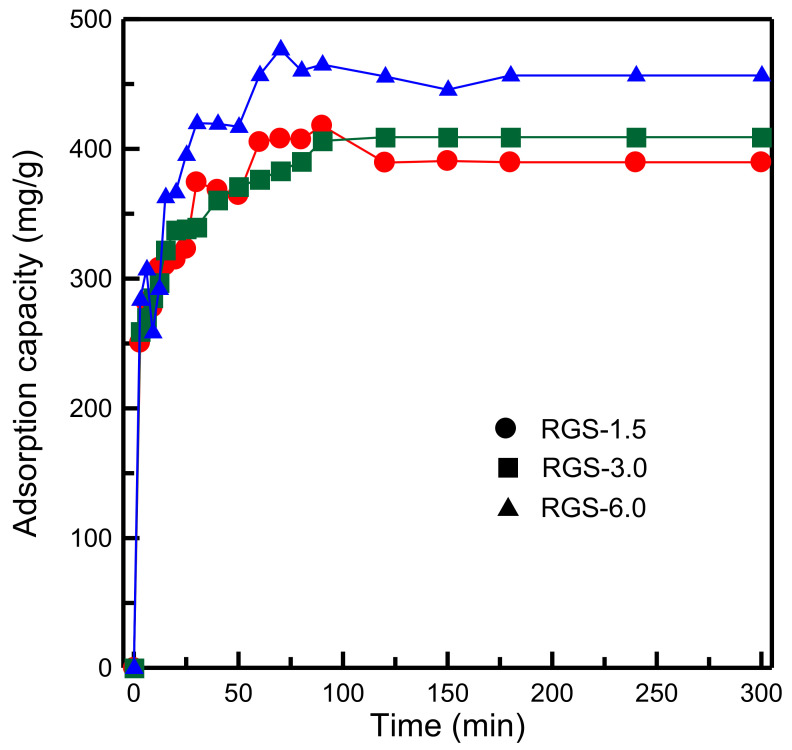
Influence of different amounts of GO suspension on adsorption capacity of RH-GO/SBA-15.

**Figure 11 materials-14-01214-f011:**
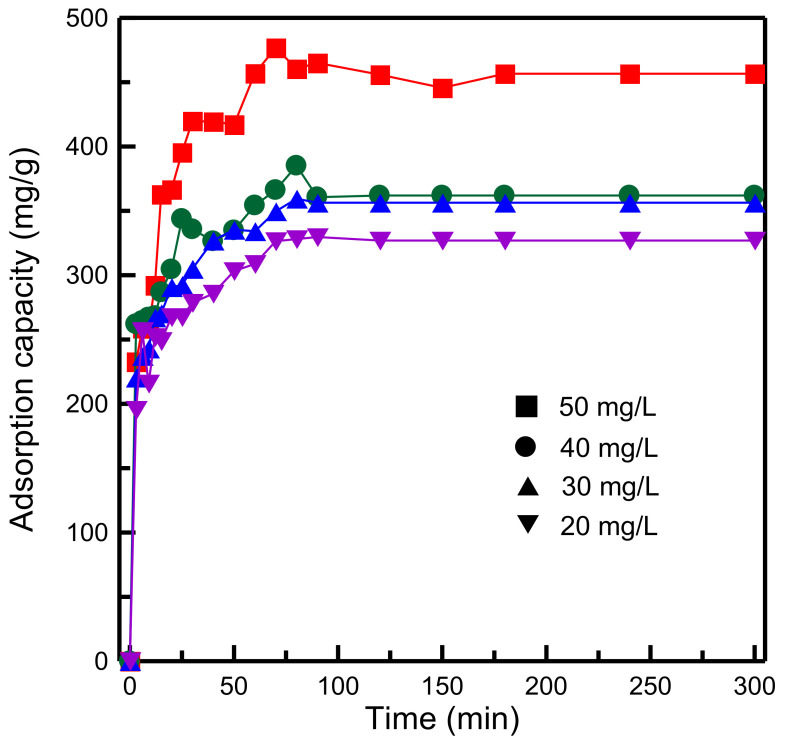
Influence of initial concentration of dye on adsorption capacity of RH-GO/SBA-15.

**Figure 12 materials-14-01214-f012:**
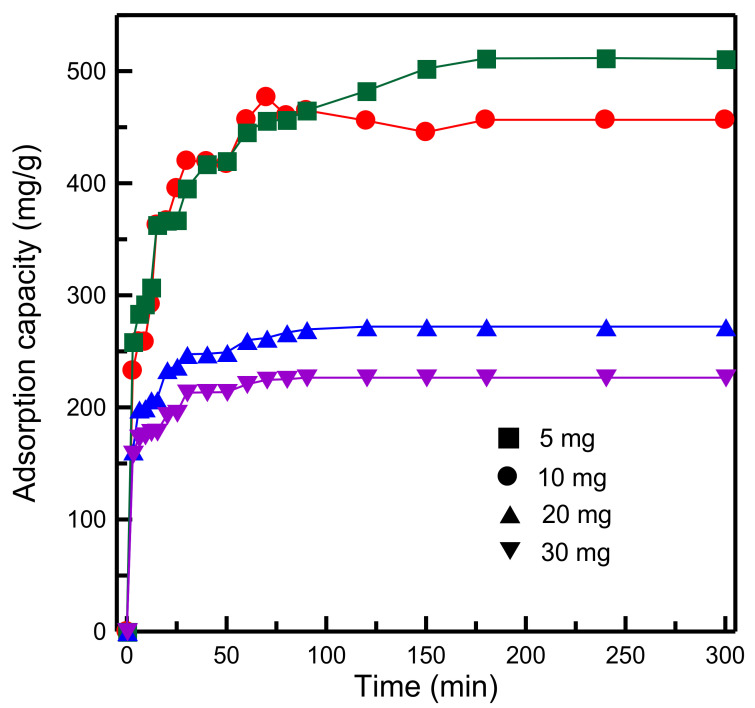
Influence of dosage of adsorbent on adsorption capacity of RH-GO/SBA-15.

**Figure 13 materials-14-01214-f013:**
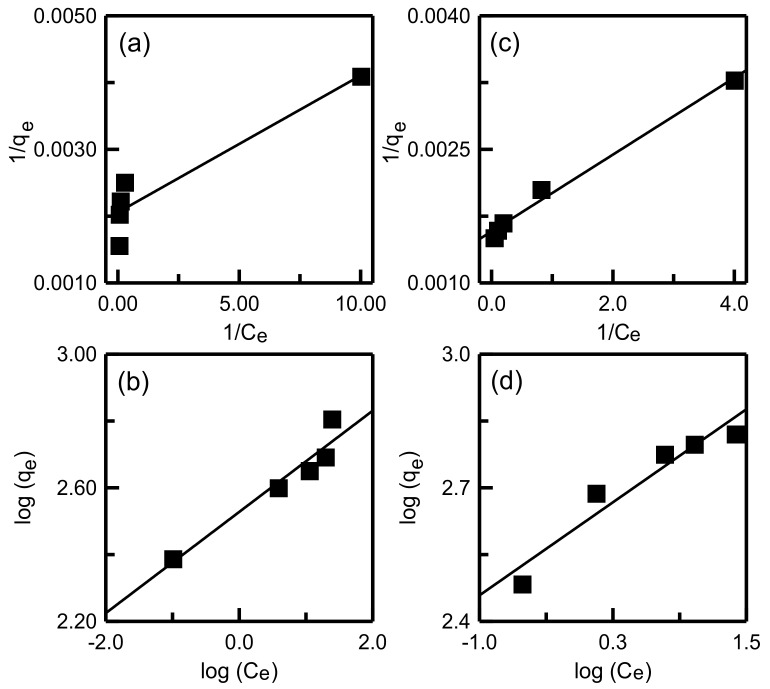
Adsorption isotherms of dye onto adsorbents: Langmuir model for (**a**) RH-SBA-15 and (**c**) RH-GO/SBA-15; Freundlich model for (**b**) RH-SBA-15 and (**d**) RH-GO/SBA-15.

**Figure 14 materials-14-01214-f014:**
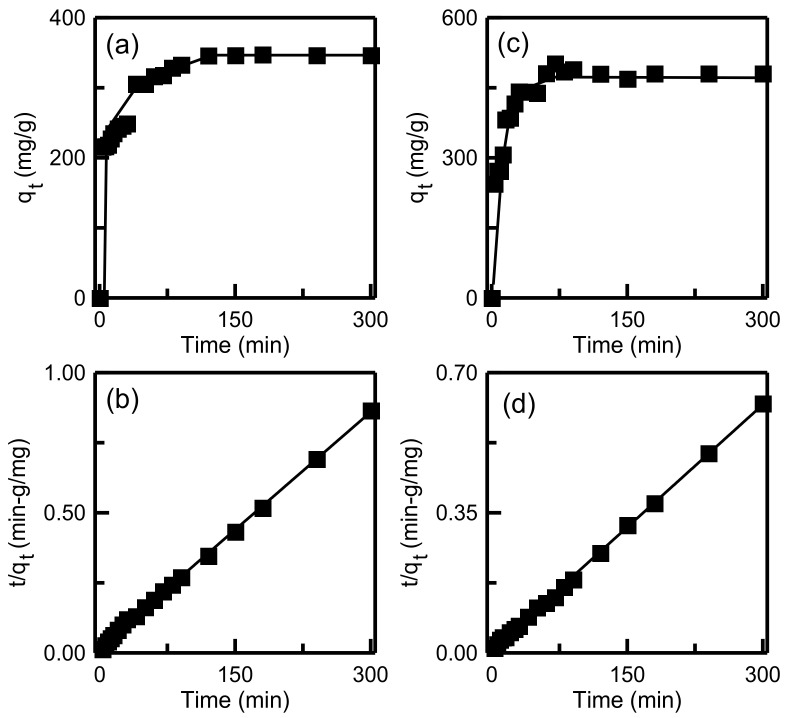
Adsorption kinetics of dye onto adsorbents: pseudo-first order model for (**a**) RH-SBA-15 and (**c**) RH-GO/SBA-15; pseudo-second order model for (**b**) RH-SBA-15 and (**d**) RH-GO/SBA-15.

**Figure 15 materials-14-01214-f015:**
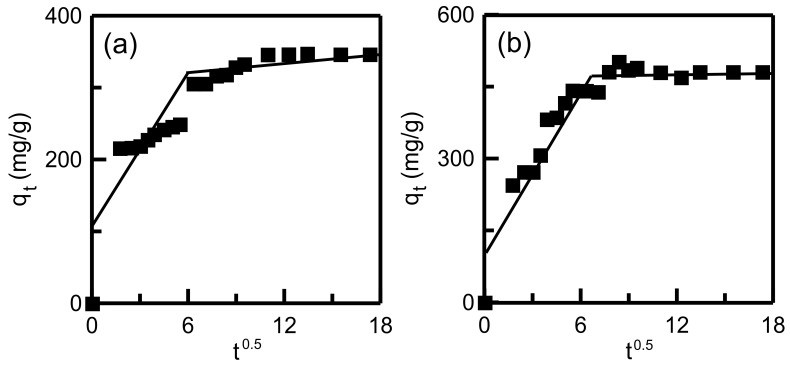
Intraparticle diffusion plot for dye adsorption onto adsorbents: (**a**) RH-SBA-15 and (**b**) RH-GO/SBA-15.

**Table 1 materials-14-01214-t001:** The intensity ratios (I_D_/I_G_) of Raman spectra for different amounts of GO suspension of adsorbent samples.

Sample	I_D-band_	I_G-band_	A (I_D_/I_G_)
RGS-1.5	8676.5	8977.9	0.9664
RGS-3.0	11,162.5	11,526.8	0.9684
RGS-4.5	13,159.4	13,565.3	0.9701
RGS-6.0	13,429.1	13,758.9	0.9760

**Table 2 materials-14-01214-t002:** Surface area, pore volume and pore diameter of adsorbent samples.

Sample	S_BET_ (m^2^/g)	V_t_ (cm^3^/g)	d_P_ (nm)
T-SBA-15	625	1.419	7.78
RH-SBA-15	656	1.092	11.73
RH-GO/SBA-15	499	0.901	11.67

S_BET_ = specific surface area, V_t_ = total pore volume, dp = pore diameter (BJH desorption).

**Table 3 materials-14-01214-t003:** Comparison of adsorption capacity for MB adsorption onto different GO-containing adsorbents.

Adsorbent	q (mg/g)	Temperature (°C)	pH	Reference
Graphene	153	25	3–10	[43]
GO	240	20	6	[44]
GO/AC	147	25–75	1.5–12	[18]
GO/chitosan	276	25	6	[45]
GO-montmorillonite/sodium alginate	151	30–60	2.28–9.70	[46]
Carboxylated GO/Carboxymethyl cellulose	180	25–55	2–12	[47]
RH-GO/SBA-15	511	25	7	Present work

**Table 4 materials-14-01214-t004:** Isotherm parameters for dye adsorption onto two types of adsorbents.

Adsorbent	Langmuir				Freundlich		
	*R_L_*	*q_L_* (mg/g)	*K_L_*	*R* ^2^	*n*	*K_F_* (mg/g)	*R* ^2^
RH-SBA-15	0.0019	478.47	10.3773	0.85892	6.83	334.37	0.92944
RH-GO/SBA-15	0.0054	632.91	3.6659	0.98485	6.18	422.22	0.89322

**Table 5 materials-14-01214-t005:** Kinetics parameters for dye adsorption onto two types of adsorbents.

Model	Parameter		Value
		RH-SBA-15	RH-GO/SBA-15
Pseudo-first-order adsorption kinetic	*q_e,experiment_* (mg/g)	346.56	481.18
	*q_e,calculated_* (mg/g)	388.82	476.05
	*k*_1_ (min^−1^)	0.1803	0.0752
	*R* ^2^	0.8145	0.9276
Pseudo-second-order adsorption kinetic	*q_e,experiment_* (mg/g)	346.56	481.18
	*q_e,calculated_* (mg/g)	295.86	490.20
	*k*_2_ (min^−1^)	0.00054	0.00822
	*R* ^2^	0.9999	0.9991
Intraparticle diffusion kinetic	*k_i_*_1_ (mg/g min^1/2^)	86.44	56.04
	*C* _1_	24.42	92.79
	*k_i_*_2_ (mg/g min^1/2^)	4.67	1.12
	*C* _2_	280.15	471.93

## Data Availability

Data sharing not applicable.

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
