# Peer review of "Utilization of Rice Husk Ash in the Preparation of Graphene-Oxide-Based Mesoporous Nanocomposites with Excellent Adsorption Performance"

_materials, 2021, doi:10.3390/ma14051214_

Round 1

Reviewer 1 Report

The submitted manuscript shows a new type of composite materials containing graphite oxide (produced by the modified Hummers method) deposited on mesoporous SiO2 obtained using rice husk ash as the silica source. These structures can be useful in adsorption of cationic dyes (such as methylene blue). The synthesized materials were widely characterized and tested in the adsorption process under various conditions, including type and dosage of adsorbent and initial dye concentration. The work seems interesting but requires revision as suggested in the list below.

  1. Why the belief that “biomass is more environmentally friendly than fossil fuels” (the first sentence of Introduction)? Of course, biomass is a renewable raw material and from this point of view more attractive than fossil fuels, but its combustion may have even worse effects than burning of, for example, natural gas.
  2. Do Authors have any idea for the application of the carbon part that remains after silica extraction from the rice husk ash?
  3. The diffractograms presented in Figure 1 should be supplemented with the XRD pattern for the material synthesized without the addition of GO, as well as the material based on the use of TEOS as the silica source (T-SBA-15).
  4. At this point I would also like to ask what was the reason that GO was added during the SBA-15 structure formation by condensation. It was not expected that GO would be incorporated into the structure of silica? Maybe it would be better to introduce GO after the formation of SBA-15 structure? Especially since, as the TEM images show, GO deposition occurred mainly outside the SBA-15 particles.
  5. The adsorption isotherms of the materials obtained using sodium silicate (from rice husk ash) are completely different compared to T-SBA-15. It is clear that we are dealing with another porosity. Can Authors then say with 100% certainty that they generated the SBA-15 structure?
  6. Surprisingly, there is no microporosity in the synthesized SBA-15 materials (see Table 2). How do Authors explain that? The classic structure of SBA-15 has a large number of microporous connectors communicating adjacent mesopores.
  7. It would be valuable to determine the real role of the individual components of the obtained composites in the MB adsorption process, if an additional curve representing the adsorption capacity for pure GO appeared in Figure 7.
  8. What is the role of pH in the studied adsorption process? Was it kept constant throughout the adsorption test?

Reviewer 2 Report

This manuscript by Liou & Liou aims to assess the use of rice husk ash as a source of silica for the synthesis of composite materials for the adsorption of methylene blue. The authors had already produced the same type of material in their previous paper and tested the adsorption potential of close material for methylene blue. Also, several papers in the literature already demonstrated the ability of rice husk (and rice husk ash) to be used as adsorbent, especially for methylene blue (e.g. Sharma et al. 2010, Desalination).

The authors must clearly write the proper novelty of this study as this manuscript looks like hundreds of other already published on this subject.

The adsorption section must be reinforced. For example you did not clearly write the pH used during experiments even if you demonstrated in your last work that the pH value strongly impact the adsorption capacity? Why did  you use kinetic experiments in order to demonstrate the impact of several parameters (gelation pH, etc.) instead of adsorption isotherms?

The adsorption isotherms with only 5 points are not enough. You must perform correct adsorption isotherm experiments in which the saturation is reached whatever the experiments.

"The results could be useful for the development of high-efficiency adsorbents to be employed in the purification of dye wastewater." To be honest, I don't precisely understand how you may reach such conclusion based on your results. For example, why did you use such complicated chemical process for the synthesis of GO/SBA materials although rice husk may be far more simple with close efficiency?

In my view, this manuscript must be strongly modified and some experiments should be performed again. Therefore, i recommend to reject this manuscript.

Reviewer 3 Report

The paper entitled "Utilization of rice husk ash in the preparation of graphene-oxide-based mesoporous nanocomposites with excellent adsorption performance" by Liou and Liou has been submitted for its consideration and possible publication in MDPI Materials. 

The use of a natural source SBA-15 makes the study interesting for the revalorization of waste, and its good performance in adsorption assays (of MB) 

The characterization includes Raman, FTIR, XRD, Electron Microscopy and BET. I find missing surface charge analysis of the prepared materials (by Zeta Potential) so this fact can help further to explain the behavior of the composite system. 

I find curious that even the green principle of using rice husk, the authors did not evaluate the direct use of graphite composite (instead of preparing GO). This may be cleared up or discussed in the manuscript. 

I would recommend publication with minor revisions, after the authors consider these two issues. 

Round 2

Reviewer 1 Report

In general, Authors have taken my suggestions into account and revised the manuscript, but the lack of microporosity in the structure of the SBA-15 materials remains unclear. I do not dispute the fact that SBA-15 silica is a mesoporous material. However, I am surprised by the micropore volume values for the synthesized materials, which in Table 2 assume the value of 0.000 cm3/g. As I wrote in my previous report, all SBA-15 materials have microporous linkers between adjacent mesopores that give a non-zero micropore volume. Therefore, I am asking to recalculate these values once again, or to explain this deviation from the standard materials described in the literature.

Reviewer 2 Report

The authors have performed some insightful comments based on the reports of the reviewers. Even if i'm still not convinced by the overuse of kinetic experiments instead of adsorption equilibrium experiments in this manuscript, this manuscript can be considered suitable for publication in Materials.
